# Contrast Sets for Evaluating Language-Guided Robot Policies

**Abrar Anwar***, **Rohan Gupta***, **Jesse Thomason**
University of Southern California
{abraranw, rgupta96, jessetho}@usc.edu

**Abstract:** Robot evaluations in language-guided, real world settings are time-consuming and often sample only a small space of potential instructions across complex scenes. In this work, we introduce contrast sets for robotics as an approach to make small, but specific, perturbations to otherwise independent, identically distributed (i.i.d.) test instances. We investigate the relationship between experimenter effort to carry out an evaluation and the resulting estimated test performance as well as the insights that can be drawn from performance on perturbed instances. We use the relative performance change of different contrast set perturbations to characterize policies at reduced experimenter effort in both a simulated manipulation task and a physical robot vision-and-language navigation task. We encourage the use of contrast set evaluations as a more informative alternative to small scale, i.i.d. demonstrations on physical robots, and as a scalable alternative to industry-scale real world evaluations.

**Keywords:** Evaluation, Language-guided Robots

## 1 Introduction

Language can be used for providing guidance on tasks like high-level task planning [1, 2], robot manipulation [3, 4], and visual navigation [5]. Robots are deployed in environments with many objects, and the space of language commands a robot can execute grows combinatorially with scene complexity. As such, these robots are often trained on large datasets that can specify hundreds of tasks in different environments. Evaluating such a robot on the large domain of its training set is impractical, especially as one typically needs to evaluate various policies. In simulation, researchers are able to evaluate their language-guided policies on robust, i.i.d. evaluation sets. However, since evaluation on physical robots is difficult, and experimenters usually *demonstrate* a robot's capabilities on a small subset of tasks, falling short of the i.i.d. evaluation framework typical in simulation. In this work, we focus on evaluating language-guided robot policies efficiently so that experimenters can explore the large space of possible instructions with less work.

Simulation is commonly used to evaluate language-guided policies [6, 7, 8, 5]. After training a policy on various tasks, the policy is evaluated on a large number of predefined test instances. Since simulations are typically insufficient for truly understanding a policy's real-world performance [9, 10], and despite correlations between simulation and reality [11], there is a need for an evaluation framework to systematically evaluate language-guided robot policies in the physical world. Since the space of possible robot behaviors in different scenes is large, these approaches must also be efficient with respect to experimenter effort.

Consider manipulation and navigation tasks where a robot follows natural language instructions. To evaluate a manipulation task, an experimenter has to move tabletop items to modify a scene, which takes experimenter effort. In navigation tasks, evaluations are trickier because the environment itself should vary between instances. Changing the environment in navigational settings often means moving furniture or adding new large objects, which is labor-intensive. Additionally, when language is involved, these objects must also be semantically relevant. As a consequence, it is difficult to evaluate the performance of a language-guided policy at scale due to experimenter effort.

---

*These authors contributed equally to this work.

8th Conference on Robot Learning (CoRL 2024), Munich, Germany.

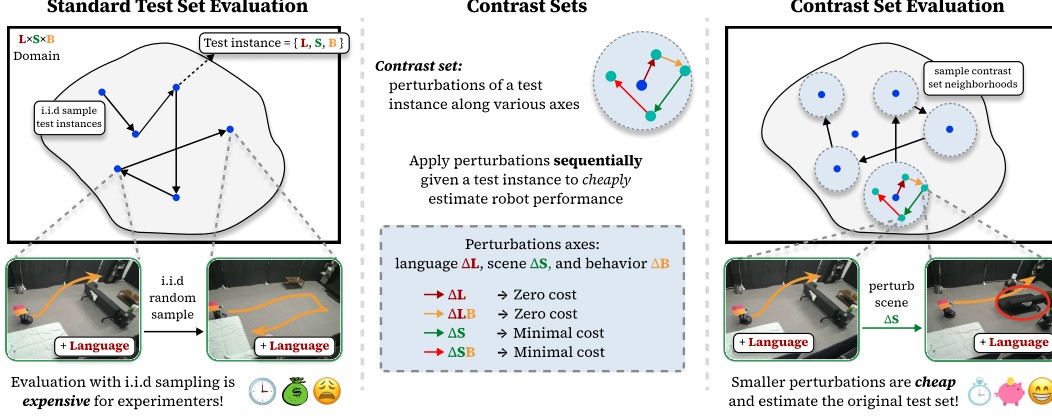

Figure 1: **Overview.** *Left:* In standard test set evaluation, a test set is i.i.d. random sampled to cover the domain of possible language, scene and behaviors that a robot can execute. It can be expensive to reset the scene to each new test instance during experiments. *Middle:* In this work, we design contrast sets [12] for language-guided robot evaluation, comprising perturbation strategies based on the language, scene, and expected behavior of the robot. *Right:* The proposed contrast set evaluation allows experimenters to efficiently evaluate neighborhoods around original test instances.

Contrast sets [12] use deliberate perturbations of i.i.d. test sets to enable better evaluation of a machine learning model. We take inspiration from contrast sets and propose perturbation functions to systematically and efficiently evaluate the space of robot test instances. In this work, we build a framework for contrast set-based perturbation strategies for evaluating language-guided robot policies. We apply perturbations on test set instances in the language, scene, and behavior axes, which allows for a lower cost evaluation. This framework strikes a balance between expensive i.i.d. sampling of test instances and cheaper test instance perturbations. In particular, we

- design contrast set-based perturbation strategies for exploring the test domain;
- demonstrate how policy performance across different types of perturbations lend insight into axes of policy robustness and brittleness;
- design a simulated manipulation task and a real-world vision-and-language navigation task and demonstrate that our contrast set evaluation efficiently estimates policy performance.

## 2 Background and Related Work

Past work in machine learning model evaluation has used perturbations as a method to probe model performance; however, evaluation in robotics typically focuses on a small number of pre-defined tasks. In this section, we discuss contrast sets from NLP, and its relevance to evaluation for robotics.

**Evaluation in Machine Learning.** A large, sampled i.i.d. test set may not capture the span of expected situations a machine learning model could encounter in the real world. To address this, researchers have designed out-of-distribution evaluation techniques in the vision [13, 14, 15, 16] and NLP [17, 18, 12] communities. In computer vision, perturbations of images have been used to generate counterfactual examples to test a model [19, 20, 21, 22, 23]. These approaches allow experimenters to stress test their models and have confidence in their model during deployment. However, in robotics, testing requires physical deployment and takes considerable efforts to compute such metrics. In this work, we focus on designing contrast sets for language-guided robot policies.

**Evaluation of Robot Policies.** Simulation is a common way to train and evaluate robot policies [24, 25, 26]. These simulated environments are often used for evaluating the performance of a real-robot system [10, 9, 27, 28] by recreating a simulated counterpart to a real environment, but show ineffective direct sim2real performance without domain randomization or real-world fine-tuning strategies. There exist correlations between simulation and real-world performance even if they do not exactly match [29, 11]; however there are no guarantees about real-world performance. These works also pre-define a set number of tasks in simulation; but it is not scalable to engineer simulators for every new task. Other recent work consider how to evaluate in language-guided robots in the real world such as evaluating LLM-based task planners [30] or providing bounds on policy

performance [31]. Additionally, RL frameworks such as autonomous RL [32] and others [33, 34] focus on minimizing the number of human interventions during deployment. Contemporary work have begun to investigate how different visual or linguistic attributes impact evaluation performance [35, 36, 37] and efficient data collection [38]. We find our work complementary to these, and focus on evaluating the space of possible test instructions and scenes in an efficient manner.

**Contrast sets in NLP.** Contrast sets are perturbed variants of the test set that help characterize the decision boundary of a classification model. They are constructed by perturbing the input text and/or the output label. For example, in a sentiment classification task, a perturbation to test model robustness to sarcasm could append "Yeah, right!" to the text of a positive review, indicating a change from a positive label to a negative label. We design contrast sets for robotics that alter accompanying language instructions of test instances and potentially the expected behavior.

**Contrast sets with vision.** In the NVLR2 [39] visual reasoning task, contrast sets are formed by replacing an image with one that contains a minimal change that may alter the answer to an accompanying question. For example, for a test that asks whether an image contains "two chow dogs facing each other", an image perturbation finds one that is semantically similar—two dogs are facing each other—but the dogs are of different breeds, thus the label is flipped. We design contrast sets for robotics that alter the start state of instances and potentially the expected behavior.

## 3 Problem Statement and Notation

Language-guided policies can be learned from large-scale collected data, then deployed to control physical robots. Systematic, efficient approaches to evaluate learned policies controlling physical robots will facilitate understanding how effectively those policies can cover the domain of test instances. In this paper, we introduce an evaluation strategy inspired by contrast sets [12] to estimate a given policy's performance on a fixed evaluation set measured by a given metric while minimizing the physical cost of setting the initial conditions of each evaluation instance.

The space of discrete language instructions is notoriously large, so these policies are typically evaluated in simulation over thousands of instructions. Several works have focused on correlating simulation performance to real world performance on a handful of instructions for tuning the sim2real gap [11] or on evaluating image-based navigation policies in Airbnbs or rented homes [40, 28]. Tailoring simulations for new skills or renting dozens of environments are not scalable paradigms as experimenters must spend dozens of hours to cover situations policies may encounter.

**Evaluation sets.** We formalize the problem of evaluating language-guided robot policies in a domain of test instances $\mathcal{X}$ and range of expected behaviors $\mathcal{Y}$. An evaluation set $X \times Y; X \subset \mathcal{X}, Y \subset \mathcal{Y}$ is composed of instances, the initial conditions $(l, s) \in X$ faced by a policy and the desired outcome $b \in Y$, characterized by the language instruction $l$, the starting scene $s$, and the expected optimal behavior $b$. For notational simplicity, we specify a language-guided policy $f$ which takes in initial conditions $l, s$ and produces behaviors $b$ rather than that of an iterated state-to-action definition: $f(l, s) \to b$. To evaluate a language-guided robot policy $f$, experimenters sample an evaluation set $X = \{X_1, ..., X_n | X_i \sim \mathcal{X}\}$ with associated behaviors $Y = \{\text{human}(X_i) = Y_i; X_i \in X\}$. We assume that the sampled evaluation set $X$ is representative of the test manifold defined by $\mathcal{X}$. A trained, language-guided robot policy $f : \theta \times X \to Y$ is evaluated over $X$ to produce $\hat{Y}$, against which a performance metric $M(\hat{Y}, Y)$ is calculated.

**Evaluation Strategies.** An evaluation strategy sequentially samples or modifies test instances $X$ into a sequence of test instances to be executed, with experimenter intervention to set up each subsequent starting condition. A standard evaluation strategy $\mathcal{I}(X) = (x_i \in X | \forall i \in 1, ..., n)$ simply converts the evaluation set into a shuffled sequence of test instances. Let $\mathcal{P}$ be a set of perturbations functions. A strategy $\Gamma$ can construct a larger test set by applying perturbations $\Gamma(X) = (\delta(x) | x_i \in X, \delta \in \mathcal{P}, \forall i \in 1, ..., n)$. In this paper, we highlight the comparative advantages of such a perturbation-based evaluation strategy.

**Cost and metrics.** To calculate the cost of a series of evaluations, we define an evaluation cost $C(\mathcal{I}(X))$. This cost will give us insight to how much experimenter effort a given evaluation strategy takes. The purpose of a test set is to estimate the value of a given metric $M(f(\mathcal{I}(X)), Y)$ that is an indicator of the robot's performance. The metric $M$ and evaluation cost $C$ are general terms and are chosen depending on the setting. The metric could be reward in an environment, success rate,

success-weighted path length, or the distance from goal, while the evaluation cost metric could be distance of objects moved, time taken, or energy used to reset the scene.

The i.i.d. random sampling strategy $\mathcal{I}(X)$ is typically ideal for estimating an evaluation metric for a given policy. However, the cost over this strategy $C(\mathcal{I}(X))$ likely exceeds any given cost budget $K$, because i.i.d. sampling of test instances is not aware of the cost of changing instances. The goal of our work is to design a strategy $\Gamma(X)$ over the evaluation set using cost-effective perturbations $\mathcal{P}$ such that $C(\Gamma_K(X)) < C(\mathcal{I}(X))$ while $M(\Gamma_K(X)) \approx M(\mathcal{I}(X))$, where $\Gamma_K(X)$ is the sequence of evaluations bounded by the cost budget $K$. This formulation means that the size of the contrast set is often greater than the size of the standard evaluation $|\Gamma_K(X)| > |\mathcal{I}(X)|$. The procedure for applying a set of perturbations $\mathcal{P}$ given a cost budget $K$ is shown in Algorithm 1 in the Appendix.

In this work, we consider the problem of efficiently evaluating language-guided robots across different instructions, scenes and behaviors. As such, we do not focus on language-conditioned pick and place tasks as the scope of perturbations is limited in these environments. We consider settings where it is possible to compose language instructions to define innumerable numbers of robot behaviors. Specifically, we use the LanguageTable [41] and VLN-CE [42] tasks, as the language instructions in these environments specifically relate to *how* the actions are taken.

## 4  Contrast Set Evaluation Strategies

We adapt contrast set perturbations to design a strategy $\Gamma(X)$ that perturbs an evaluation set in the context of language-guided, visual sequence-to-sequence problems. We then instantiate these perturbations in a simulated manipulation task and a physical vision-and-language navigation task.

**Contrast sets for robots.** Each test instance is characterized by the language instruction $l$, the scene $s$, and the expected behavior $b$ (Figure 1). Perturbations to the language and scene can lead to changes in the expected behavior, so we define our own scene and language perturbation functions that may or may not modify the expected behavior. We define four types of perturbation functions, denoted by the symbol $\Delta$ and a letter for the axes they modify.

- $\Delta L(x)$ perturbs the language instruction such that the expected behavior is the same.
- $\Delta LB(x)$ perturbs the language instruction such that the expected behavior is different.
- $\Delta S(x)$ perturbs the environment such that the expected behavior is the same.
- $\Delta SB(x)$ perturbs the environment such that the expected behavior is different.

Instantiated perturbation functions $\mathcal{P}$ depend on the tasks being evaluated. We measure cost $C$ with respect to modifying the environment, so language-based perturbations do not add any additional physical cost. We do not include the cost of resetting the robot position as it is fixed across evaluations, and in some cases can be automated. Our work investigates whether contrast sets are able to estimate the *sample* mean performance of a standard evaluation set with a similar or lower cost.

**Contrast sets vs i.i.d. evaluation sets.** In comparison to a standard evaluation $\mathcal{I}(X)$, where an experimenter sequentially executes i.i.d. random test instances with no consideration of the cost, contrast set evaluation allows the experimenter to explore the test domain and cover it efficiently.

Often, when researchers *demonstrate* the capabilities of their policy, they do not construct an i.i.d. evaluation set, but instead only apply perturbations. These demonstrations are typically done to cheaply evaluate their robot in a limited domain. However, because those test instances and perturbations are not i.i.d., those works are not properly evaluating their robot systems. Since our work assumes access to an i.i.d. test set and then applies perturbations in sequence, contrast sets for robots strikes a balance between saving experimenter effort and properly evaluating a robot policy. Through more systematic evaluations in the test domain, contrast sets enable better coverage of the test domain, accurate performance estimates, and insights on a policy's sensitivity to perturbations.

## 5  Language-Table Simulator Experiments

Language-Table [41] is a multi-task language-conditioned control benchmark that spans 696 unique task conditions across five categories, with each specified in language in dozens of ways. The general task is for a manipulator to push blocks of various shapes and colors to specific relative or absolute positions based on a language instruction, as visualized in Figure 2. It is infeasible to evaluate all of

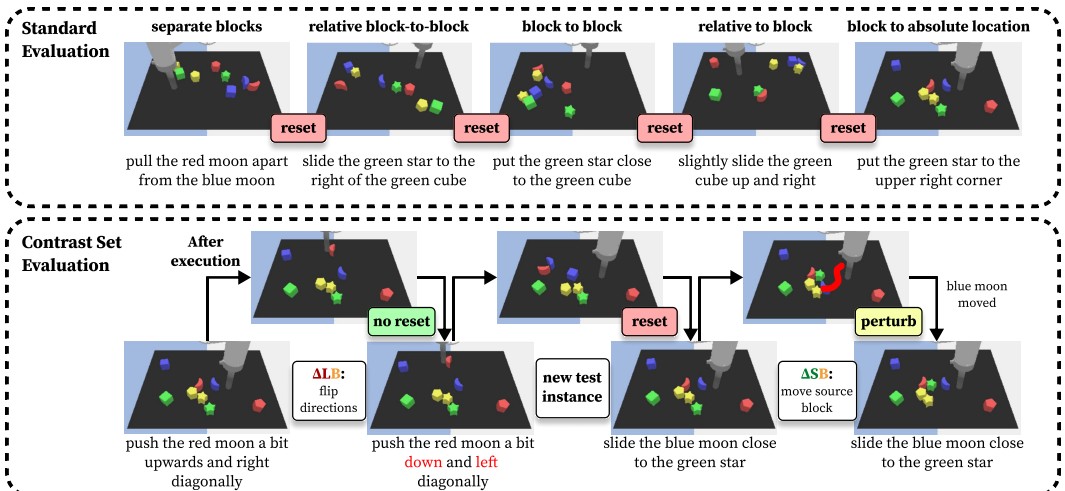

Figure 2: **Language-Table Rollouts.** In the Language-Table simualtor [41], we sample an evaluation set of 250 test instances that is sequentially evaluated. A test instance is sampled from one of five task types which manipulate blocks according to a task definition. The standard evaluation requires i.i.d. random sampling instructions and scenes, which accumulate more effort for the experimenter. Contrast set evaluation allows experimenters to perturb sampled test instances by making minimal changes after each execution, leading to less work for the experimenter.

these test instances in a physical environment, especially if there are multiple checkpoints or models to evaluate, making it important to study how to efficiently evaluate large task domains.

## 5.1 Experiment Design

We compare a standard evaluation set $X$ against strategies $\Gamma(X)$ that use various perturbation sets $\mathcal{P}$. Over three seeds, we sample an evaluation set $X$ from the simulator consisting of 250 test instances sampled across five different task categories, from which we run various evaluation strategies.

**Perturbations.** Using perturbation functions $\mathcal{P}$, we perturb these test instances with low cost. We believe it would normally be infeasible to reset the scene after each evaluation, so between perturbed test instances, we do not reset the environment. However, if we applied perturbations that do not modify the expected goal of the robot, a perturbed test instance may already be in a success state. Therefore, we do not use the $\Delta L$ and $\Delta B$ functions and define two $\Delta LB$ perturbations and two $\Delta SB$ perturbations as shown in Figure 2. The test instructions typically involve a source block that needs to be moved, a target block that a block needs to move relative to, or a direction such as "top" or "right". The $\Delta LB_1$ perturbation swaps the target and source block referring expressions in the instruction. $\Delta LB_2$ modifies the instruction such that any directions or positions are flipped, for example *upper left* is changed to *lower right*. $\Delta SB_1$ moves the target block to a new location, while $\Delta SB_2$ moves the source block to a new location. Given a pre-defined test set, we design various contrast sets from these perturbation strategies to determine their impact on effort.

**Metrics.** We use an optimal planner to compute the the success weighted by path length (SPL) as the metric $M(\cdot)$ such that suboptimal trajectories are penalized. As a proxy for experimenter effort in resetting a scene, we define the cost $C(\cdot)$ as the distance (in meters) objects are moved in a scene.

## 5.2 Results and Discussion

Figure 3 summarizes the differences in policy performance on contrast sets that target particular language- and scene-based perturbations versus the original test set and shows that contrast set evaluation does not compromise the estimate of the test set $M(\cdot)$ achieved via Standard Evaluation.

**Contrast sets show the policy's sensitivity to perturbations.** Since the perturbations are concrete and specific, they can show a policy's sensitivity to the language and scene axes. As shown in Figure 3, we find that the language-based perturbation $\Delta LB_2$ is notably below the mean SPL for the

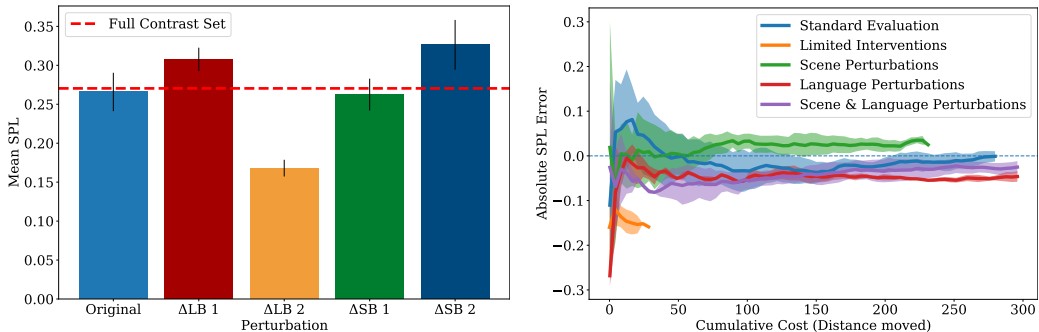

Figure 3: **Left:** A key insight offered by contrast set evaluations is probing the strengths and weaknesses of a learned policy. The mean success-weighted path length (SPL) achieved over the full test set may compare average policy performances, but here we observe additional robustness to instruction source and target switches and source block starting position ($\Delta LB_1$, $\Delta SB_2$) but brittleness to direction word inversion ($\Delta LB_2$), providing insights for training and deployment. **Right:** Comparison of evaluation strategies' absolute estimation error of the SPL of the entire test set as a function of the cumulative cost in distance blocks are moved during scene resets. The maximum cost of the standard evaluation is 281, achieving the horizontal error line at 0.0, and we cap cost at 300, though additional perturbation instances are possible for some strategies. All perturbation strategies achieve better test set SPL estimates than a Limited Intervention baseline.

policy. $\Delta LB_2$ simply swaps the directions in text to be the opposite. We took a test instruction of moving a block to the right changed the direction to left with $\Delta LB_2$, and we found that the policy would continue to move the block to the right This reduction in performance indicates that the trained model may be overfit to direction in some cases. By contrast, $\Delta SB_2$'s higher SPL relative to the contrast set indicates the policy is robust to source block locations. Intuitively, this result shows that perturbations help qualitatively characterize different regions of the test manifold.

**Limited interventions are not predictive of SPL.** Standard evaluation i.i.d. samples new test instances. However, in practice, sometimes experimenters simply execute new language instructions in a given scene until they cannot anymore, which allows them to reduce the amount of effort required by minimally intervening to reset their scene. We explore a limited intervention evaluation strategy in which new language instructions are sampled from the evaluation set sequentially without resetting the scene. This experiment is meant to be a lower bound in cost, where new language instructions and behaviors are sampled, but the reset costs are is minimized by not resetting the scene unless a planner determines an instruction is infeasible. As shown in Figure 3, 250 trials of this strategy yields much lower cost than all of the other strategies, but poorly estimates the test SPL.

**Contrast sets improve predictive performance and provide a cost-effective approach to executing more trials.** Figure 3 shows that language-based perturbations underestimate the SPL of the evaluation set while scene-based perturbations overestimates it. Using all four perturbation functions together yields the best predictive performance, as it converges closest to the full evaluation set performance. As shown in Figure 6 in the Appendix, the standard evaluation incurs a cost of 281 meters over 250 trials. In comparison, the language-based contrast set reaches around 400 trials for the same cost but under-estimates test performance. The combined scene and language-based contrast set allows for 380 trials without significant under- or over-estimation of the test performance.

## 6  VLN-CE Evaluation on a Physical Robot

Vision-and-language navigation in continuous environments [42] (VLN-CE) is a task where an agent follows fine-grained language instructions in a household setting. We deploy a VLN-CE agent to control a real world Locobot [43]. By evaluating with contrast sets, we are able to draw insights about policy sensitivity and estimate the full test performance with a lower cost to the experimenter.

### 6.1  Experiment Design

We design a pseudo studio apartment environment populated with furniture similar to categories in simulation. To ensure ecological validity of test instances, we recruited five participants to design

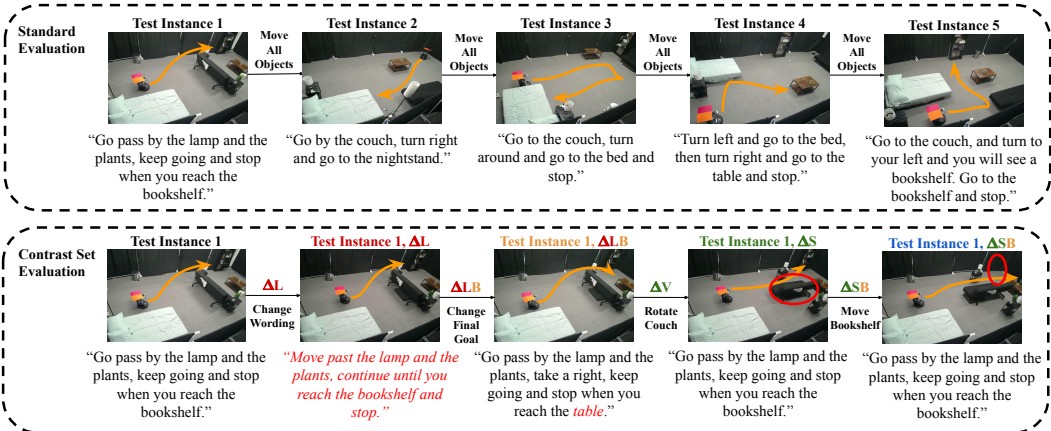

Figure 4: **VLN-CE Robot Rollouts.** The standard evaluation of i.i.d. random sampling scenes requires scenes to be shuffled around drastically. Intuitively, contrast sets allow experimenters to cheaply perturb sampled test instances to find new ecologically valid samples to evaluate.

five furniture setups, each of which they annotated with a navigation instruction as shown in Figure 4. Each test instance $x$ and outcome $y$ is defined by the scene $s$ based on the furniture arrangement and the robot's pose, and the expected behavior $b$ of the robot, described by the language instruction $l$ consisting of two subgoals to be reached. More details on this protocol for ensuring ecological validity and model training can be found in Appendix C.

**Perturbations.** We define four perturbation functions as shown in Figure 4. We define language-only perturbations $\Delta L$ to simply change the wording of the instruction but preserving the meaning. If a language instruction tells the policy to go to a bed then to the couch, then this perturbation simply rephrases the instruction. $\Delta LB$ changes the final goal in the text instruction, so the first half of the robot's behavior to the first goal is the same, but the latter half changes. $\Delta S$ moves an object around in the scene such that the expected trajectory of the robot is still the same. For example, passive objects the robot is not meant to interact with may change positions as long as they do not change the interpretation of either language subgoal. Lastly, $\Delta SB$ moves an object around in the scene such that the expected trajectory of the robot is different. This perturbation either moves the goal object such that the trajectory must be different, or an object is moved in front of the robot to impede its originally-intended trajectory.

**Metrics.** We define the cost metric $C$ as the distance objects were moved in the scene. The metric $M$ is the average progress toward two subgoals: 50% for reaching the first and 100% for reaching both. We evaluate each test instance three times. Four perturbation strategies applied to five original test instances yield 20 new instances Executing these 20 test instances and the original 5 test instances three times results in a strategy set size of $|\Gamma(X)| = 75$.

## 6.2  Results and Discussion

Modifying the scene, robot malfunctions, and maintenance all add to the cost of evaluating robots. We find that contrast sets reduce the effort required for evaluating robots in the real world.

**Contrast set perturbations enable more experiments for less work.** The contrast set consists of 4 perturbation functions. Based on Figure 8 in the Appendix, the average cost per instance in the standard evaluation set is 8.8 meters, while generating 4 perturbed trials from a single i.i.d. test instance costs about 1 meter. With 5 i.i.d. test instances from the standard evaluation set, the additional cost of 1 meter from perturbations allows one to execute $4 * 5 = 20$ more trials.

**Contrast sets can estimate the full test set with less cost.** Figure 5 demonstrates that contrast set evaluation converges to a similar success rate as standard evaluation while requiring less effort, indicating that this evaluation allows experimenters to efficiently measure a policy's performance.

**Contrast sets can provide insights into a policy's linguistic and visual understanding.** Similar to results in the simulation results, our specific perturbations can show a policy's sensitivity to specific

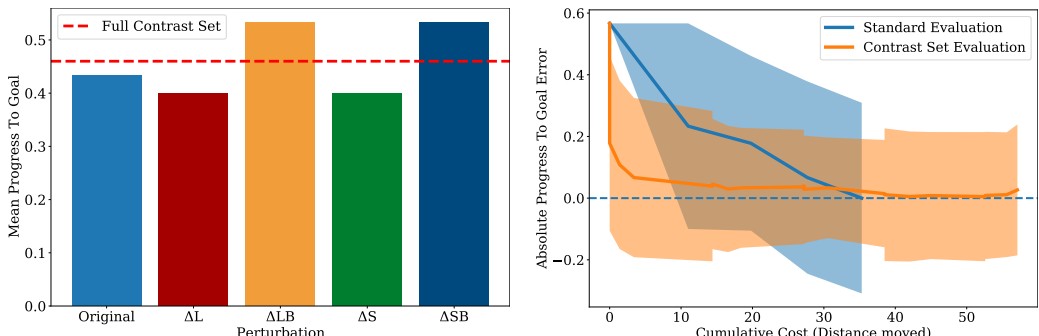

Figure 5: **Left:** Contrast set evaluation probes the strengths of a trained VLN-CE model to a physical robot. We observe that the policy is robust to changes to the final goal instruction ($\Delta LB$) and physical changes to the goal itself ($\Delta SB$) as depicted by the 13% higher performance over the full contrast set. **Right:** Average cumulative progress to goal error (and cumulative std. dev.) vs. cumulative cost. The contrast set evaluation quickly reaches a nearly accurate estimate of the final test set progress to goal, showcasing the potential to reduce experimenter costs dramatically by exploring neighborhoods of contrast around test instances.

axes. We find that $\Delta L$ and $\Delta S$ is lower in performance than that of the full contrast set, especially compared to the higher performing contrast sets. For example, we found that instruction "Go pass by the lamp and the plants, keep going and stop when you reach the bookshelf" performed with an average progress to goal of 100% over three runs, while the reworded version in the same setting achieved 33%. We found this trend to continue, and it directly highlights potential issues in the language encoder, as the language representations are likely not well aligned. Though our perturbations for this experiment are relatively general, more specific perturbations can highlight regions of the test domain where a policy is particularly effective and where it may need improvement.

# 7 Conclusion and Limitations

A language-guided robot can be tasked with potentially thousands of tasks across arbitrary environments. To evaluate such a robot, experimenters ideally construct i.i.d. test sets that effectively cover the domain of possible test instances. However, creating these sets often require extensive rearrangement of the environment, increasing experimenter effort. In this work, we propose contrast sets as an approach for evaluating language-guided robot policies. We find that evaluating these contrast sets provides insights into policy robustness and sensitivity. Additionally, contrast sets are able to estimate the full evaluation set performance while maintaining low experimenter cost. We conducted simulated manipulation and real-world vision-and-language navigation experiments and found that contrast sets enable experimenters to run more evaluations with less effort. We argue that contrast set evaluations offer higher fidelity than small-scale, real robot demonstrations while not requiring the industry-level resources for large-scale, deployed evaluations.

While we focused primarily on environments where language instructions specify greater details on how the robot should interact with the environment, we intend to investigate larger manipulation-focused datasets such as DROID [44] and OXE [45]. Additionally, we want to emphasize ecologically valid language instructions that perturbations may deviate from, which could involve new scene perturbations (e.g., new object instances), language perturbations (e.g., changing cultural contexts), or different cost types (e.g., labor costs). We assumed the existence of a test set that is representative of the test manifold, and that contrast set perturbations will generate instances within the domain. Unlike other works, we do not aim to estimate performance outside this domain. Typically, test sets are not well-defined in advance and future work can explore which perturbations can allow experimenters to efficiently evaluate more open-ended robot systems. Now that our work has shown the utility of perturbation strategies for efficiently evaluating robot policies, future work can consider automatic approaches to selecting which perturbations to apply and for generating new test instances for experimenters, potentially using large language models.

## Acknowledgments

This work was supported in part by a grant from the Army Research Lab (ARL) Army AI Innovations Institute (A2I2), award number W911NF-23-2-0010. Rohan Gupta was supported by a USC Provost's Undergrad Research Fellowship during a portion of this work. The authors would like to thank Elle Szabo for building the early infrastructure for the physical robot experiments.

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

# A    Contrast Sets for Robotics

Overall, our work investigates whether contrast sets are able to estimate the *sample* mean performance of a standard evaluation set with a similar or lower cost. We do not make estimates of the true population mean of the performance (i.e. run the experiment a large number of times so that the performance converges to a single true mean). Instead, we focus on sample mean performance comparisons because experimenters can realistically only estimate the sample mean and not the true population mean.

---

**Algorithm 1** Contrast set with perturbations

---

1: **Input:** Test set $X$, budget $K$, perturbation functions $\mathcal{P}$
2: $c_{total} \leftarrow 0$
3: **for** $x \in X$ **do**
4:     **for** $\delta \in \mathcal{P}$ **do**
5:         $c \leftarrow \text{cost}(x', x)$
6:         $x', \hat{y} \leftarrow f(\delta(x))$                                                    $\triangleright$ $x'$ is the resultant state
7:         $c_{total} \leftarrow c_{total} + c$
8:         **if** $c_{total} > K$ **then**
9:             **End**
10:        **end if**
11:    **end for**
12: **end for**

---

# B    Language-Table Simulator Experiments

We describe additional details left out of the main text on the Language-Table simulator.

## B.1    Model Details

For the policy, we use the pretrained FiLM-conditioned ResNet architecture that was trained using behavior cloning provided by the Language-Table repository [41]. We do not use Language-Table's LAVA model as a pretrained model was not provided and requires 64 TPUv3 chips to train.

## B.2    Additional Details

In this section, we describe how the cumulative cost plot in Figure 3 was generated. Since we evaluated over three seeds and each experiment has a different cost, we create 50 bins at equal intervals from 0 to the max overall cost across all seeds, then aggregate the cumulative absolute SPL error and cumulative cost. Using this binning approach, we also compute the standard deviation of the error bounds.

## B.3    Additional Results

**Contrast sets allow for more evaluations with less cost.** As depicted in Figure 6, the slopes of each type of perturbations determines how the cost scales compared to the number of evaluations. Limited interventions is clearly the lowest cost; however, we had found that it does not estimate the evaluation set. All contrast set strategies have a higher slope than that of standard evaluation. For example, scene and language perturbations can execute nearly double the number of experiments compared to the standard evaluation given a cost budget of 281.

# C    VLN-CE Evaluation on a Physical Robot

We use a Locobot [43] robot to run vision-and-language navigation in continuous environments [42] (VLN-CE) in the real world.

## C.1    Model Details

We pretrain a policy for the robot on the VLN-CE task in the Habitat simulator using the RxR training set [46]. We then use a behavior cloning objective to finetune the simulation-trained model on a small set of real world examples using teacher-forcing. The policy uses a discrete action space of forward, rotate left by 30 degrees, rotate right by 30 degrees, and stop. Only one scene arrangement was used in the training dataset, and this scene was not used during testing. We note that the furniture, especially larger items such as beds and couches, were used during training and existed during training. However, the scene arrangements, which is key to the task of VLN-CE, was ensured to be different.

## C.2 Experiment Design

We describe how we collected our test instances. We design a pseudo studio apartment environment which is populated with furniture similar to those found in simulation. To ensure ecological validity of test instances, we recruited five participants to design five furniture setups. They were instructed to ensure that the

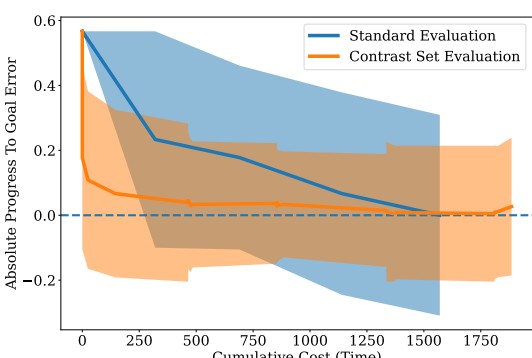

Figure 7: Average cumulative progress to goal vs cumulative cost as *time to perturb the scene in seconds*. We find that the results found in Figure 3 and Section 6.2 when using time as the cost function instead of the distance objects were moved still hold when switching to time to perturb the scene as the cost. This shows that we may be able to use various cost metrics to measure experimenter effort.

furniture was arranged in any way they would prefer, defining the scene $s$. They then placed the robot and walked a trajectory $b$ they wanted the robot to execute while narrating a natural language command $l$. A subset of the navigation instructions can be found in Figure 4. By using external participants to design our test instances, we hope to ensure that we, as experimenters, do not bias the collection of our test instances to be easier than expected.

## C.3 Additional Results

**Contrast sets allow for more evaluations with less cost.** As depicted in Figure 8, the slopes of each type of perturbations determines how the cost scales compared to the number of evaluations. Though contrast set evaluation has a higher bound, given a cost budget of 35, contrast sets allow a user to run nearly triple the number of trials for the same cost budget.

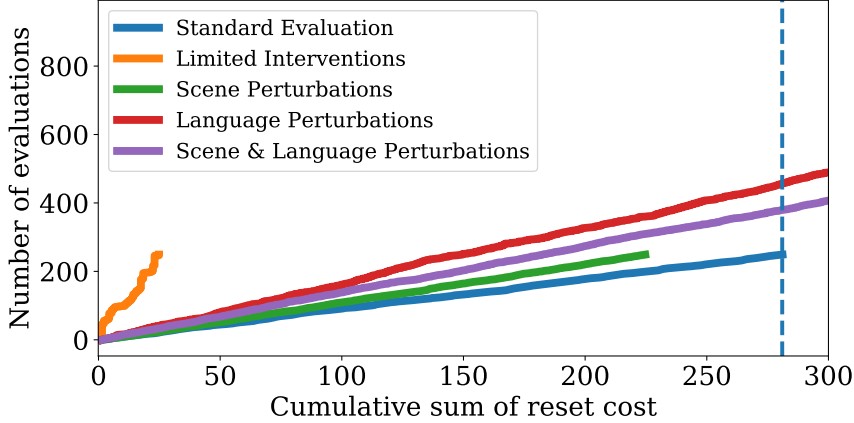

Figure 6: Compared to Figure 3, we separate the relationship between cost and error. Limited interventions and language-only perturbations allows for more evaluations with less cost, and standard evaluation has the least number of evaluations for the cost. As described in the main text, scene and language pertubations finds a good middle ground with more evaluations for less cost.

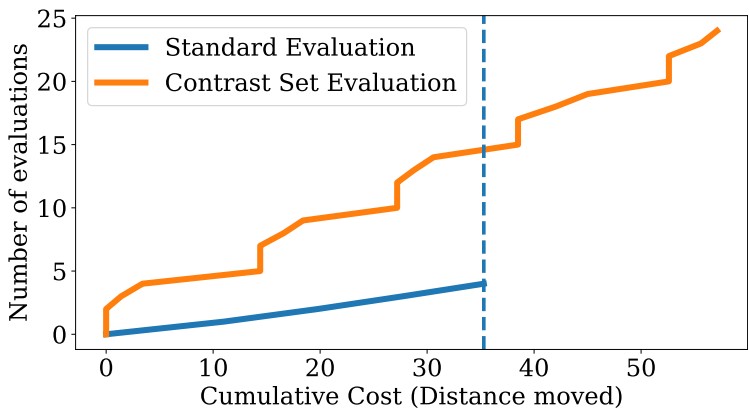

Figure 8: We separate the relationship between cost and error in the real word VLN-CE experiments. We note that the number of evaluations performed in the standard evaluation is relatively low compared to the contrast set evaluation. Contrast set evaluation allows the experimenter to execute more experiments compared to standard evaluation. Though not every single experiment from the test set can be executed under a cost budget of around 35 (blue dotted line), Figure 5 indicates that contrast set evaluation still estimates the test set.

**Contrast sets also estimate the full test set while minimizing time to reset the scene**. Instead of using distance of objects moved during a scene reset as we did in the main text, we also investigate the time used to reset the scene as a cost metric. We find similar results in Figure 7 which uses time as cost as we did in Figure 5 which uses distance of objects moved as cost. This is likely due to the nearly-linear relationship between time it takes to move items in the scene and the distance they are moved.

