# OpenReview forum: "Contrast Sets for Evaluating Language-Guided Robot Policies"
_robot-learning.org/CoRL/2024/Conference — CoRL 2024_

### Official Review · Reviewer_DoyM · 2024-07-20
**Good and timely idea, but the experimental execution and paper presentation needs some improvement.**

**Originality:** 3
**Technical Quality:** 2
**Clarity Of Presentation:** 2
**Potential Impact:** 3
**Recommendation:** 3
**Confidence:** 4

**Review:**

Strengths:
* The paper topic is pertinent to the research community as real-world on-robot evaluations are expensive, while simulation does not yet match trends on robot hardware.
* The literature review is thorough and does a good job contextualizing the paper in the literature.
* The intuitive discussion of the different perturbations and insights for the simulated experiments were helpful.
* The figures generally illustrate the main ideas and high-level procedures in the paper.
* It's great that a good number of test instances (250) is used across 5 categories and 3 seeds.
* The discussion regarding the size of the contrast set and the resulting performance right before Sec. 6 is insightful. Could this information be illustrated in a figure?

Weaknesses:
* I found the exact methods, contributions, and experiments to be a bit confusing to understand.
    * By the end of the introduction, I was still not entirely clear on what exactly was being proposed. In the abstract, it's not really clear what is meant by 'use contrast sets to characterize policies at reduced experimenter effort' - this becomes clear later in the paper, but some more context would be helpful earlier on.
    * The exact form of the inputs and outputs of the policy and the initial conditions should be specified in the 'Evaluation sets' section.
    * By Sec. 4, there was still a lot of repetition of the ideas, but not much substantial information on **how** the contrast sets achieve efficient evaluation.
    * It's also confusing what exactly the proposed procedure is. My understanding is a certain number of evaluations are predetermined based on an effort/cost budget. Then perturbations are applied to each one of these predetermined evaluation configurations. Since there are no resets (see comment below), it seems like these perturbations are just applied sequentially? But this would mean that the standard evaluation set is not a subset of the contrast set as stated in line 256. Overall, it appear the perturbations double size of the original set - but it would be helpful if the authors confirmed this. Regardless, the clarity of these points needs to be improved in the paper. If my understanding is correct, it should come across that: the standard evaluation set is chosen according to a budget, and then the contrast set is supposed to double the size, while making it reflective of a much number of evaluations. This last point is not discussed, so I am not sure if I am understanding correctly.
    * In the experiments, the standard evaluation (pre-perturbation) is compared to the perturbed dataset. It seems to me that in order to show that the proposed contrast sets achieve efficient yet reflective evaluation, they need to be compared to a much larger evaluation set. E.g., 400 evaluations for contrastive sets and 1000 evaluation for standard set/oracle baseline. Please correct me if my understanding is wrong.
    * I do not understand the 'limited interventions' setting. What is the difference between this setting and standard evaluation? My understanding is the number of evaluations is the same, but the former induces fewer changes or less cost? I would want to see a balanced set of fewer evaluations in comparison to the standard evaluation and the contrast sets. Would this smaller balanced sampling also still achieve reflective performance or it would be to stochastic/not representative? This is also a more likely evaluation strategy for an experimenter with limited resources.
* (Minor) It's probably a good idea to add a few other simulators to the related works (e.g., Robomimic, RoboCasa, RLBench).
* (Minor) There is a sentence that says object-based manipulation is not considered because there are not that many options, but one of the experimental settings appears to be object-based manipulation?
* I found it strange that no resets were considered. In my experience, robot evaluators reset the scene between evaluations. More support would be helpful for this design choice. Are there available citations that avoid resets?
* It seems that the paper is presenting a way to perturb the initial conditions and remain reflective of performance without perturbations in terms of generalization vs failure. However, the paper needs to endow the reader with an understanding of how contrast sets work and present some insights from this analysis which I found a bit lacking.
* The discussion right before the conclusion should be more specific to/grounded in the insights from the experiments run on the hardware.


* The following is a non-exhaustive list of typos and grammatic errors. The paper should be proofread thoroughly.
1. Line 92: 'have have'.
2. Line 94: 'or evaluated' is awkward/incorrect grammar.
3. Line 107: 'a sequences'.
4. Line 117: 'is chosen' should be in the plural form.
5. Line 145-146: 'this is cost is fixed'.
6. Line 156: 'contrast sets also finds'.
7. Fig. 2 caption: 'leading less work'.
8. Line 191: 'sensitivity different perturbations'.
9. Line 194: 'SPL' is not defined yet in the text (defined later in caption of Fig. 3).
10. Line 210: 'Using the distance blocks are moved in the scene as a proxy'.
11. Line 222: 'in continuous environments simulation task'.
12. Line 250: Should the size not be 20 x 3 = 60?
13. Line 255: 'more enable'.
14. Lines 281-284: 'We find...' sentence is grammatically awkward and quite long.
15. Make sure to capitalize acronyms in the paper titles of the references appropriately.

**Quality Of The Limitations Section:**

2

**Questions For Rebuttal:**

In addition to the weaknesses listed above, I have the following questions for the authors.
1. Why did you choose to perturb the initial state and the language? Are there other options? Would they be worth considering?
2. Do you have a reference of a paper that evaluated policies on thousands of instructions? Although that is the goal of embodied foundation models, I am not sure that that has really been achieved yet. If not, I would just reframe these statements as - we are going towards this evaluation scheme.
3. Is the mean in Fig. 3 over the unperturbed evaluation set? If so, why is it above the 'Original' bar plot?
4. What is the number of standard evaluations that could be achieved with 1 m of cost (w.r.t. the discussion in lines 255-258)?
5. How can the cost of the contrast set be less than the cost of the standard evaluation when the former is a superset of the latter?

**Robotics Focus:**

4

**Summary Of Paper:**

The paper presents an approach to efficient real-world robot evaluation using contrast sets to perturb a pre-determined set of evaluation criteria (in terms of state and language input). The perturbed evaluations should be reflective of larger scale evaluations, while optimizing for minimum experimenter effort. Experiments are done on a simulated manipulation task and a real-world evaluation task.

**Summary Of Recommendation:**

Overall, the paper presents a good idea towards efficient on-robot evaluation. However, the presentation of the method and experiments needs further clarity. The experiments also need to be expanded to compare standard evaluation with many more evaluations than the contrast set method and a baseline that does a balanced standard evaluation with fewer evaluations than the contrast set method. With these improvements, the paper will be a good contribution to the robot learning community.

---

### Official Review · Reviewer_uVPW · 2024-07-21
**Review of Contrast Sets for Evaluating Language-Guided Robot Policies**

**Originality:** 3
**Technical Quality:** 3
**Clarity Of Presentation:** 4
**Potential Impact:** 3
**Recommendation:** 3
**Confidence:** 4

**Review:**

Strengths:
- This work makes an effort to formalize evaluation of robot policies and to make evaluations cheaper, which is extremely important to improve time between experiments and iterations.
- The evaluations find that the performance on certain perturbations will be underestimated by iid evaluation while other perturbations are overestimated. This is an interesting finding as it implies certain perturbations create harder evaluation scenarios than others and can have implications on how we should be training policies.
- This work studies model sensitivity to language perturbations, which is somewhat rare in recent robotics works.
- Paper is written clearly.

Weaknesses:
- The experiments evaluate policies trained on narrower datasets than what recent works train and evaluate on. It would be interesting to evaluate existing pre-trained models such as Octo or DROID to see if similar observations hold.
- It’s not clear how these findings might generalize to a wider range of scene perturbations. Currently, the experiments consider different starting positions, but there may be other relevant perturbations such as object instances/shapes.

Some relevant citations:
- This is a recent work [1] that considers the cost of setting up different environments/scenarios but for data collection, which would be of interest to the authors.
- There are also some works that study sensitivity to different perturbations [2, 3, 4].

[1] Gao et al. Efficient Data Collection for Robotic Manipulation via Compositional Generalization. 2024.

[2] Xie et al. Decomposing the generalization gap in imitation learning for visual robotic manipulation. 2023.

[3] Jin et al. Video Transformers under Occlusion: How Physics and Background Attributes Impact Large Models for Robotic Manipulation. 2024.

[4] Pumacay et al. The colosseum: A benchmark for evaluating generalization for robotic manipulation. 2024.

**Quality Of The Limitations Section:**

3

**Questions For Rebuttal:**

- What are other types of scene and language perturbations that this evaluation framework can capture?
- Do contrast sets enable more efficient evaluations of existing large pre-trained models like DROID and OXE/Octo?
- Defining the cost for changes in the object positions is natural because the space is continuous. How should cost be measured for discrete perturbations like swapping a new object instance into the scene?

**Robotics Focus:**

4

**Summary Of Paper:**

This paper proposes a new way to perform robot evaluations for language instruction-based tasks. Specifically, it looks at language and scene perturbations, and explicitly consider the cost of making these evaluations. The goal is make evaluations of robot policies more accurate and cheaper.

**Summary Of Recommendation:**

Overall, I appreciate this paper’s effort in finding cheap proxies for performance of robot policies. While the scope is limited and specific, the proposed contrast sets would likely still be useful when a wider class of perturbations is considered.

---

### Official Review · Reviewer_5QZG · 2024-07-21
**Review of 524**

**Originality:** 3
**Technical Quality:** 3
**Clarity Of Presentation:** 4
**Potential Impact:** 3
**Recommendation:** 3
**Confidence:** 3

**Review:**

This paper is a nice breath of fresh air amidst the constant incremental algorithmic advances in language-guided robot learning today, offering instead to critically re-evaluate how evaluation is conducted in testbed domains.

*Strengths*:
* The paper is well-written and well-motivated, offering a straightforward and intuitive solution to what is an often swept-under-the-table question, that of critical evaluation of language-based policies.
* The definition of contrast sets for evaluation is well-grounded in adjacent learning-based fields, and make reasonable sense in a language-conditioned robotics context.

*Weaknesses*:
* While the authors offer four reasonable contrast set-based perturbations (lines 139-142), this list feels still a small subset of what is likely the true representative set of concern for full-scope evaluation. For example, any linguist in the field knows that human intent and pragmatics are immensely important to language-based task specifications and may often vary wildly across different human users (e.g., people who use the robot can ground on different context depending on preference and thereby produce utterances that are different from each other that are not influenced by the environment). How do the authors intend on differentiating between preference-based language perturbations that are inconsistent between human users?
* The evaluation domains are quite contrived. For example, I am somewhat suspect that fully descriptive commands such as "“Go pass by the lamp and the plants, keep going and stop when you reach the bookshelf" is actually what a human user would ever utter as a fully-explicated task specification to a robot (they're far more likely to say something like "go to the bookshelf". Can the authors justify the veracity of human language commands like these wrt to natural human utterances for language commands given?

**Quality Of The Limitations Section:**

2

**Questions For Rebuttal:**

* How did the authors come up with the 4 specific perturbations to be tested, and what is the justification/reasoning for why they believe this set to be comprehensive of the full space of perturbations that may exist in language-based evaluation?
* As stated above, it's very unlikely that real human language specifications will be as highly descriptive as the two domains tested. How do the authors imagine their method to extend or not extend in real-world domains with far more ambiguous linguistic commands?

**Robotics Focus:**

4

**Summary Of Paper:**

This paper contributes a novel way to construct robot test sets by using contrast-based perturbation strategies to design test instances.

**Summary Of Recommendation:**

This paper provides a reasonable advancement to evaluation suites for language-guided robots. While I think it requires more technical delineation to be practically feasible, I would still be in favor of accepting the paper for its explicit focus on evaluation, an area that I feel is oft ignored in learning-based robotics.

---

### Author Rebuttal · Authors · 2024-08-13

Thank you to the reviewers and area chair for their feedback. We are encouraged by how our work was received and that reviewers found our work as important. We created a PDF draft of the primary changes we made based on the reviewers’ feedback. The red text contains the changes. We will polish and compress the text in the final version of the paper, but here we summarize the changes.

In each individual response, we discuss the relevant changes in detail. Our primary focus based on the reviewer’s feedback was to improve clarity, citations, scaling to other environments, and ecological validity of language instructions. We did this by:

- introducing details on contrast sets earlier into the paper
- adding an algorithmic procedure to explicitly show how contrast test instances are generated
- introducing terminology on sample and population means to make our claims more clear
- discussing the role of the limited interventions strategy
- making references to various results we had in the appendix
- improving clarity of the cost of standard evaluations in the real world
- adding examples of the insights contrast sets allow for in the real world
- including citations on autonomous RL, other simulators, other datasets, and efficient data collection methods
- adding mentions to ecological validity in our tasks
- adding note on future work on preference-based language, types of perturbations, and types of cost
- fixing typos and small grammatical errors

In the final version of the paper, we plan on adding the additional experiments proposed by Reviewer 5QZG, as we think they’ll provide new insights on how roboticists should evaluate their policies and for how long. We believe we have addressed all concerns given by the reviewers, but if the reviewers have any other suggestions, please let us know!

---

### Decision · Program_Chairs · 2024-09-04

**Decision:**

Accept

**Comment:**

The paper presents a framework for real-world robot evaluation using contrast sets to perturb a pre-determined set of evaluation criteria and explicitly considers the cost of making these evaluations to optimize for minimum experimenter effort.

Strengths: The reviewers agree on the importance of the problem setup.

Weaknesses: The main criticism lies in the scalability of the proposed approach to wider data distributions and models and improving the clarity of the presentation.

Post response period: The authors have addressed some of reviewer's concerns. In particular, they have improved the overall clarity of the presentation. However, adding more comprehensive experimental results in more environments/datasets would help make the paper stronger. Given the author response, the updated ratings and their justifications, the AC recommends the paper to be accepted with a poster presentation.